# Nipple Sparing Mastectomy as a Risk-Reducing Procedure for *BRCA*-Mutated Patients

**DOI:** 10.3390/genes12020253

**Published:** 2021-02-10

**Authors:** Nicola Rocco, Giacomo Montagna, Carmen Criscitiello, Maurizio Bruno Nava, Francesca Privitera, Wafa Taher, Antonio Gloria, Giuseppe Catanuto

**Affiliations:** 1G.RE.T.A. Group for Reconstructive and Therapeutic Advancements, 80100 Naples, Italy; 2Breast Service, Department of Surgery, Memorial Sloan Kettering Cancer Center, New York, NY 10001, USA; montagng@mskcc.org; 3Breast Center, University Hospital of Basel, 4056 Basel, Switzerland; 4Department of Oncology and Haematology (DIPO), University of Milan, 20122 Milan, Italy; carmen.criscitiello@ieo.it; 5Division of Early Drug Development for Innovative Therapy, European Institute of Oncology, IRCCS, 20122 Milan, Italy; 6G.RE.T.A. Group for Reconstructive and Therapeutic Advancements, 20122 Milan, Italy; maurizio.nava@gmail.com; 7Multidisciplinary Breast Unit, Azienda Ospedaliera Cannizzaro, 95021 Catania, Italy; f.privitera05@gmail.com (F.P.); giuseppecatanuto@gmail.com (G.C.); 8Surrey and Sussex Health Care, NHS Trust, London W1W 5EF, UK; misswtaher@yahoo.com; 9Institute of Polymers, Composites and Biomaterials—National Research Council of Italy, 80125 Naples, Italy; antonio.gloria@cnr.it; 10G.RE.T.A. Group for Reconstructive and Therapeutic Advancements, 95021 Catania, Italy

**Keywords:** breast cancer, *BRCA* mutations, risk reduction, nipple-sparing mastectomies

## Abstract

Growing numbers of asymptomatic women who become aware of carrying a breast cancer gene mutation (*BRCA*) mutation are choosing to undergo risk-reducing bilateral mastectomies with immediate breast reconstruction. We reviewed the literature with the aim of assessing the oncological safety of nipple-sparing mastectomy (NSM) as a risk-reduction procedure in *BRCA*-mutated patients. Nine studies reporting on the incidence of primary breast cancer post NSM in asymptomatic *BRCA* mutated patients undergoing risk-reducing bilateral procedures met the inclusion criteria. NSM appears to be a safe option for *BRCA* mutation carriers from an oncological point of view, with low reported rates of new breast cancers, low rates of postoperative complications, and high levels of satisfaction and postoperative quality of life. However, larger multi-institutional studies with longer follow-up are needed to establish this procedure as the best surgical option in this setting.

## 1. Introduction

Fifty-five to sixty-five percent of women with *BRCA1* mutation and 45% of women with *BRCA2* mutations will develop breast cancer by the age of 70 years [1,2]. Growing numbers of asymptomatic women who become aware of carrying a *BRCA* mutation are choosing to undergo risk-reducing bilateral mastectomies with immediate breast reconstruction [3], due to the so-called Angelina Jolie effect [4]. In those cases, NSM is the preferred option, allowing better patient-reported outcomes [5,6]. Based on current evidence, risk-reducing mastectomies in *BRCA* mutations carriers reduce the risk of subsequent breast cancer by 89–95% [7]. The role of NSM in high-risk populations has seldom been reported. As NSMs have been associated with residual glandular breast tissue, in particular underneath the nipple-areolar complex [8], the oncological safety in terms of risk reduction in *BRCA* mutated patients is of particular concern. The aim of this review was to assess the literature supporting the oncological safety of NSM in *BRCA* mutated patients.

## 2. Methods

The literature review was conducted according to the PRISMA statement [9]. We searched PubMed, Cochrane Library, and Google Scholar from database inception to 31 December 2020. The following terms were used (including synonyms and closely related words) as index terms: *BRCA* mutations, NAC-sparing mastectomy, nipple-sparing mastectomy, prophylactic mastectomy, risk-reduction mastectomy.

### 2.1. Eligibility Criteria

We included all studies reporting breast cancer events after NSM was performed in asymptomatic patients carrying a pathogenic *BRCA* mutation or a variant of unknown significance (VUS).

### 2.2. Study Selection

Two investigators independently reviewed articles for eligibility based on the study titles and abstracts, and studies that met the inclusion criteria were retrieved for full-text assessment, data extraction, and inclusion in the review. All disagreements were resolved by consensus.

### 2.3. Data Collection Process

The included studies were screened by two independent authors (NR and GM), who evaluated the study design and main results. Data extraction was performed using a standardized tool. A level of evidence, according to the Oxford Centre for Evidence-Based Medicine [10], was assigned by the same two authors. Discrepancies were resolved by consensus. To evaluate the methodological quality of the studies selected, the Newcastle-Ottawa Quality Assessment Scale (NOQAS) for case-control studies [11] was used. The methodological quality of the case series was evaluated using a modified version of the standardized tool developed by Murad and colleagues [12], excluding drug-related items (4, 5, and 6). A modified version of the cohort studies Newcastle-Ottawa Quality Assessment Scale was used to evaluate cohort studies excluding items (2 and 5) related to non-exposed subjects. Quality scales were implemented by two independent investigators (NR and GM) to evaluate the overall quality of each eligible study better and discuss all discrepancies in scoring. Due to the different ranges of the included quality scores, results were normalized to a range between 0 and 10. We considered a normalized-score of ≥7 as high quality.

## 3. Results

Nine studies met the inclusion criteria and were included in this review [13,14,15,16,17,18,19,20,21] (Figure 1). Two studies, from the same institution considered the same group of patients [20,21], and, therefore, only the one with longer follow-up [21] was included. Two studies were assessed as (Level of Evidence) LoE III and six studies as LoE IV, while the quality of the studies ranged from 5 to 7 (Table 1).

Jakub et al. [19] retrospectively reviewed the outcomes of nine institutions’ experience with risk-reducing NSM from 1968 to 2013 in a cohort of patients with *BRCA* mutations. Bilateral prophylactic NSMs were performed in 202 patients. The median age at the time of surgery was 41.5 years. After prophylactic NSM, no breast cancers developed in the nipple-areola complex (NAC), skin flaps, subcutaneous tissue, mastectomy scar, chest wall, or regional lymph nodes on the side of the risk-reducing procedure at a median follow-up of 34 and 56 months (respectively for *BRCA1* and *BRCA2* mutated patients). The authors also calculated that approximately 22 new primary breast cancer events would have been expected without risk-reducing procedures, with a significant reduction in breast cancer events (*p* < 0.001) in patients undergoing NSM.

Domchek et al. conducted a large multicenter cohort study of 2482 women with *BRCA1* or *BRCA2* mutations treated between 1974 and 2008 at 22 European and North American centers. The median age at surgery was 40.7 years. They reported that none of 247 patients who underwent prophylactic skin-sparing mastectomy (SSM) developed breast cancer during a three-year follow-up, as opposed to 7% of women without risk-reducing mastectomy who developed a breast cancer event [14].

Hartmann et al. followed a small cohort of 26 *BRCA* mutated patients, who underwent SSM at the Mayo Clinic (median age at surgery 36.5 years), and also reported that none developed breast cancer at 13 years follow-up [13].

Peled et al. reported that none of 26 patients who underwent risk-reducing NAC-sparing mastectomy at a median age of 41.2 years at the University of California San Francisco (UCSF) developed a primary breast cancer at 51 months follow-up [16].

Yao et al. reported the development of one breast cancer (outside from the preserved nipple-areolar complex) in a group of 150 *BRCA1/2* mutation carriers who underwent NSM for risk reduction (mean age at surgery: 41 years), at the NorthShore University HealthSystem (NSUHS) and Massachusetts General Hospital (MGH), at a median follow-up of 32.6 months [17].

None of the 63 patients in the study by Manning who underwent risk-reducing NAC-sparing mastectomy (median age: 39 years) at Memorial Sloan Kettering Cancer Center developed breast cancer at a follow-up of 2.5 years [18]. In a follow-up study from the same institution, Valero and colleagues reported that at a median follow-up of 36.8 months, there had been no new breast cancer diagnoses [21]. These results confirm the initial findings reported from De Alcantara Filho et al. [20].

Harness et al. (15) also reported no breast cancers in six asymptomatic *BRCA* mutation carriers who underwent risk-reducing NAC-sparing mastectomy, in a community hospital setting, at a median follow-up of 18.5 months.

## 4. Discussion

Nine retrospective studies have investigated the safety of NSM as a risk-reducing procedure in asymptomatic *BRCA* carriers. All studies, except the one from Hartmann and colleagues [13], have a relatively short follow-up (range 18.5–41.8 months). Additionally, all but two studies by Jakub and Domchek [14,19] included a small number of patients (range 20–150).

The majority of the studies also included subgroups of patients with a history of breast cancer who underwent therapeutic NSM. Although the percentage of *BRCA1/2* for the group of patients who underwent prophylactic NSM was not always reported [13,15,17,18,19]. The median age at surgery was similar across the studies (range 36.5–41 years). The incidence of primary breast cancer post-NSM in asymptomatic *BRCA* mutated patients who underwent risk-reducing bilateral procedures was low and consistent across the studies.

According to a recent systematic review, residual glandular breast tissue is reported in up to 100% of patients undergoing mastectomy and is mainly associated with the type of surgical procedure, indication, and surgeon’s expertise [8]. Residual breast tissue can be found in all areas of the remaining chest wall, mostly in the skin-flaps and more frequently underneath the nipple-areolar complex [8]. This poses a risk for breast cancer development in these patients. In a study by Papassotiropoulos, 2019 [22], residual breast tissue (RBT) was detected in 82/160 (51.3%) mastectomies. The median RBT percentage per breast was 7.1%. Of all factors considered, only the type of surgery (40.4% for SSM vs. 68.9% for NSM; *p* < 0.001) and surgeon (*p* < 0.001) were significantly associated with RBT. This evidence is particularly relevant when considering the generalizability of the results of our review, as all the included studies have been conducted in highly specialized breast centers with dedicated breast surgeons, treating a high volume of patients.

The probability of nipple involvement by premalignant lesions (i.e., ductal carcinoma in situ) in the nipple-areolar complex of *BRCA* mutation carriers is low at the time of risk-reducing procedures [23]. Manning et al. highlighted the importance of focused retro areolar tissue assessment and designation of a separate nipple margin even in *BRCA* carriers undergoing prophylactic NAC-sparing mastectomy [18]. However, the practice of focused retro areolar tissue assessment in asymptomatic *BRCA* mutation carriers undergoing risk-reducing NSM is currently not recommended by international guidelines [24,25]. According to the results of our literature review, NSM appears to be a safe option for *BRCA* mutation carriers from an oncological point of view, as the reported rates of postoperative complications are low in all the included series. However, surgical training and surgeon experience are crucial in limiting complications and the amount of breast tissue left behind.

An important aspect to consider is the impact of NSM on quality of life. Unfortunately, none of the studies in this review included data on patient-reported outcomes (PROMs). However, other studies have looked explicitly at PROMs in *BRCA* carriers who underwent risk-reducing surgery. Metcalfe and colleagues reported that women who underwent NSM have better body image and sexual functioning compared with SSM, while both groups had comparable levels of cancer-related distress and perception of breast cancer risk [26]. Keller and colleagues conducted a retrospective study of 39 *BRCA1/2* carriers, half of whom were asymptomatic at the time of surgery. The great majority (87%) underwent NSM. At a median follow-up of 5.6 years after surgery, most patients were satisfied or very satisfied with the cosmetic outcome (33/39; 85%). Only four reported discontentment (4/39; 10%) with the cosmetic outcome, and two (2/39; 5.1%) would elect a different type of operation. Improvement in quality of life was seen in 78% of patients, with a reduction in anxiety being the most important factor. None of the 39 patients reported regrets despite approximately 50% experiencing some degree of pain and a minority (7/39; 18%) reporting moderate limitations in everyday life and limitations in leisure time activities (4/39; 10.3%) [27]. Salibian and colleagues evaluated patient satisfaction in 22 very young (<30 years) *BRCA* carriers who underwent NSM and found similar results. At an average follow-up of 37 months, none of the patients regretted their decision, and 83.3% of patients would have the procedure at the same age, 8.3% at an earlier age, and 8.3% at a later age. Twelve patients also completed the BREAST-Q surveys: average scores for satisfaction with breasts and information were 73 and 77.8, respectively. Average well-being scores were 79 (Physical), 78.2 (Psychosocial), and 79.6 (Sexual) [28]. These findings are in line with a recent Cochrane meta-analysis that evaluated the efficacy of risk-reducing mastectomy and its impact on the quality of life in both *BRCA* and non-*BRCA* carriers [29]. In this view, NSM as a risk-reducing procedure for *BRCA*-mutation carriers appears to be an oncologically safe procedure, comparable to the standard modified radical mastectomy [30], with better patient-reported outcomes.

## 5. Conclusions

Nipple-sparing mastectomy appears to be an oncologically safe option for *BRCA* mutation carriers, with low reported rates of new breast cancers, low rates of postoperative complications, and high levels of satisfaction and postoperative quality of life. However, larger multi-institutional studies with longer follow-up are needed to establish this procedure as the best surgical option in this setting.

## Figures and Tables

**Figure 1 genes-12-00253-f001:**
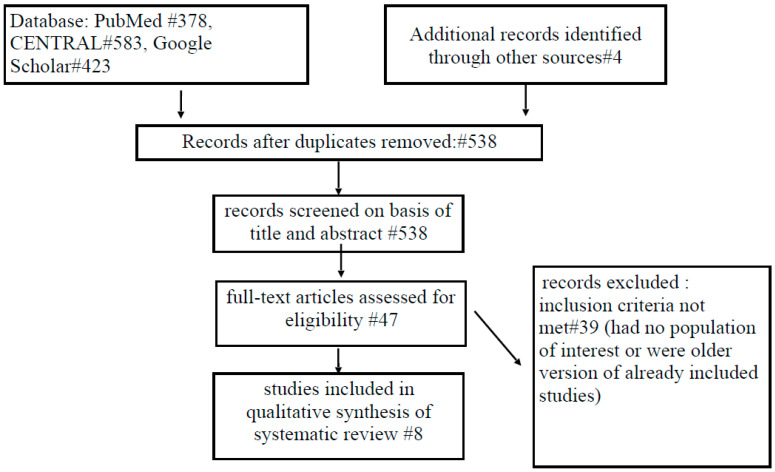
PRISMA flowchart.

**Table 1 genes-12-00253-t001:** Characteristics of included studies.

	^#^*BRCA* Mutation Carriers Undergoing Risk-Reducing NAC-Sparing Mastectomy	Breast Cancer Diagnosis Following Risk-Reducing NAC-Sparing Mastectomy	Median Follow-up (Years)	Median Age at Time of Surgery (Years)	1	2	Level of Evidence	Quality Score
Hartmann 2001 [13]	26	0	13.4	36.5 (§) 44 (§§)	NR	NR	3	7
Domchek 2010 [14]	247	0	2.5–3.7 (in different groups)	40.7	159	88	3	7
Harness 2010 [15]	6	0	1.5	NR	NR	NR	4	5
Peled 2014 [16]	26	0	4.3	41.2	14	12	4	6
Yao 2015 [17]	150	1	2.7	41 (mean)	NR ^#^	NR ^#^	4	7
Manning 2015 [18]	63	0	2.2	39	NR *	NR *	4	6
Jakub 2018 [19]	202	0	2.8 and 4.7 (in different groups)	41	NR **	NR **	4	7
Valero 2020 [21]	117	0	3	41.5	72	45	4	7

NA, not applicable; NR, not reported; §–§§, The study separately reports median age at surgery for deleterious mutations (§) and uncertain mutations (§§); ^#^ The study also included 51 patients with breast cancer who underwent therapeutic NSM (total *n* = 201). The authors reported the breast cancer gene () status for the full cohort (125 (62%) had a 1 mutation and 76 (38%) a *2* mutation). * The study also included patients with a history of breast cancer who underwent contralateral NSM (total *n* = 89). The authors reported the status for the full cohort (56 (63%) had a 1 mutation, 26 a *2* (29%) mutation, and 7 (8%) had a variant of unknown significance). ** The study included 144 patients with a history of breast cancer who underwent contralateral NSM (total *n* = 346). The authors reported the status for the full cohort (201 (58%) had a 1 mutation and 145 (42%) a *2* mutation).

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
