# Peer review of "Nipple Sparing Mastectomy as a Risk-Reducing Procedure for BRCA-Mutated Patients"

_genes, 2021, doi:10.3390/genes12020253_

Round 1
Reviewer 1 Report
This is an original systematic review of literature on nipple sparing mastectomy for mutation carriers.
The selection of the literature is well done as well as the statistical analysis.
There are some points would deserve attention/modification.
- NAC is used in many countries as acronyms of Neoadjuvant Chemotherapy. For nipple Sparing Mastectomy would be better to use NSM.
- “ Growing numbers of asymptomatic women who become aware of their 35 BRCA mutation choose to undergo risk-reducing bilateral mastectomies with 36 immediate breast reconstruction (3)” I would be interesting to mention why?
- The probability of nipple involvement by premalignant lesions in the NAC 137 of BRCA mutation carriers is low at the time of risk reduction procedure (19). 137 – 138
Not clear this definition. Not clear what are the premalignant lesions increasing the risk, if one.
Author Response
Thank you, we appreciate your positive comments. We agree with your suggestions and we have adjusted the text accordingly.
- We have used the acronym NSM consistently throughout the text.
- We have added a paragraph exploring possible reasons why BRCA carriers are increasingly choosing to undergo risk reducing surgery.
- We have defined the terms “premalignant lesions”.
Reviewer 2 Report
The present systematic review aims to assess the prognostic value of the bilateral nipple sparing mastectomies in patients with BRCA mutations. The manuscript is very interesting and simple to read, but some revisions are needed, in order to render the manuscript more complete. In this version the manuscript seems more a mini-review rather than a systematic review. Please below point by point some concerns:
1) the filters used for the search literature is missing
2) The inclusion and exclusion criteria are not reported
3) The figure 1 with the flow-chart for the inclusion of the papers, in accordance with the PRISMA method is missing. Please add and explain the reasons for the exclusion.
4) Table 1 is poor. Please insert more information about the surgery, demographical characteristics of the patients, type of tumors and other information that would improve to understand the type of articles and to critically analyse them.
5) a quality assessment of the selected studies is missing
6) some references are not reported in the discussion (see Manning et al)
7) data about the Quality of Life should be further reported. Please discuss and add information.
Author Response
Thank you for your comments. We agree with your suggestions and we have adjusted the text accordingly.
1. We did not use filters for our literature search. This is because any study design and any age group was allowed.
2. Inclusion criteria are specified in the Methods section. We have added a sentence to specify the process used for studies considering the same population.
3. We added the flow chart as for the PRISMA method, explaining reasons for exclusions.
4. We have extracted more data from the original studies and we have added them to table 1, this include the rate of BRCA1 and BRCA2 mutations, median age at the time of surgery, the level of evidence according to the Oxford Centre for EBM and the quality of the study according to the Newcastle-Ottawa Scale and Murad scale for case series. Only 1 breast cancer event was reported in the 9 included studies, unfortunately tumor subtype, size and histological characteristics of this single cancer were not available.
5. We have added a quality assessment for each of the included studies according to the Newcastle-Ottawa Quality scale and the level of evidence assessment according to the Oxford Centre for EBM.
6. We added the missing references in the discussion section
7. Unfortunately none of the included studies reported data on quality of life. Nevertheless, in the discussion section we have reported data from other studies that have assessed PROMs in BRCA carriers undergoing NSM.
Reviewer 3 Report
The authors have nicely summarized the literature pertaining to BRCA mutation and NAC-sparing mastectomy from several databases. They show that the NAC-sparing mastectomy in BRCA mutant patients is a safer option, with low cases of breast cancer and overall better post-operative quality of life for women. This review provides an important rationale based on a literature survey for choosing NAC-sparing mastectomy for women with BRCA mutation.
Author Response
Thank you. We appreciate your positive comments regarding our work.
Round 2
Reviewer 2 Report
The manuscript is significantly improved after the careful revisions.